# Compositional Generalization Across Distributional Shifts with Sparse Tree Operations

**Paul Soulos**[*]
Johns Hopkins University
psoulos1@jh.edu

**Henry Conklin**[*]
University of Edinburgh

**Mattia Opper**[*]
University of Edinburgh

**Paul Smolensky**
Johns Hopkins University and
Microsoft Research

**Jianfeng Gao**
Microsoft Research

**Roland Fernandez**
Microsoft Research

## Abstract

Neural networks continue to struggle with compositional generalization, and this issue is exacerbated by a lack of massive pre-training. One successful approach for developing neural systems which exhibit human-like compositional generalization is *hybrid* neurosymbolic techniques. However, these techniques run into the core issues that plague symbolic approaches to AI: scalability and flexibility. The reason for this failure is that at their core, hybrid neurosymbolic models perform symbolic computation and relegate the scalable and flexible neural computation to parameterizing a symbolic system. We investigate a *unified* neurosymbolic system where transformations in the network can be interpreted simultaneously as both symbolic and neural computation. We extend a unified neurosymbolic architecture called the Differentiable Tree Machine in two central ways. First, we significantly increase the model's efficiency through the use of sparse vector representations of symbolic structures. Second, we enable its application beyond the restricted set of tree2tree problems to the more general class of seq2seq problems. The improved model retains its prior generalization capabilities and, since there is a fully neural path through the network, avoids the pitfalls of other neurosymbolic techniques that elevate symbolic computation over neural computation.

## 1 Introduction

Deep learning models achieve remarkable performance across a broad range of natural language tasks [71], despite having difficulty generalizing outside of their training data, struggling with new words [36], known words in new contexts [29], and novel syntactic structures, like longer sequences with greater recursive depth [30, 39]. Increasingly this problem is addressed through data augmentation, which tries to make it less likely a model will encounter something unlike what it sees during training — reducing the degree by which it has to generalize [1, 15, 26]. However, even models trained on vast quantities of data struggle when evaluated on examples unlike those seen during training [32].

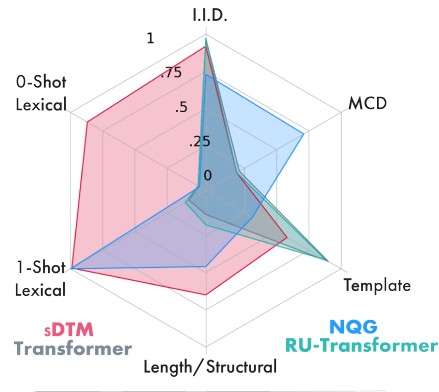

Figure 1: Generalization ability of our approach (sDTM) compared with baselines across various out-of-distribution shifts, averaged over different datasets. See §5.

---

[*]Work partially completed while at Microsoft Research.

38th Conference on Neural Information Processing Systems (NeurIPS 2024).

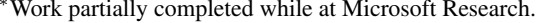

This stands in contrast to how humans process language, which enables robust generalization [51]. By breaking novel sentences into known parts, we can readily interpret phrases and constructions that we have never encountered before (e.g. 'At the airport I smiled myself an upgrade', [22]). Why do models trained on orders of magnitude more language data than a human hears in 200 lifetimes [25] still fail to acquire some of language's most essential properties?

Central to language's generalizability is compositional structure [49] where contentful units, like words, fit together in a structure, like a syntactic tree. Many classical approaches in NLP and Machine Learning attempt to induce a grammar from data in the hope of leveraging the same kinds of generalization seen in natural language [e.g. 34, 33, 68]. However, structured representations are not first-order primitives in most neural networks [44, 63]. Despite theoretical appeal, the strictures of purely discrete symbolic approaches have made them difficult to apply to the breadth of tasks and domains where deep learning models have proven successful [18]. In contrast, purely connectionist models — like Transformers [71] — struggle with the kinds of sample efficiency and robust generalization ubiquitous to human learning.

Neurosymbolic methods attempt to integrate neural and symbolic techniques to arrive at a system that is both compositional and flexible [4, 19, 20, 63]. While some neurosymbolic architectures achieve impressive compositional generalization, they are often brittle due to the symbolic core of their computation [58]. These methods are hybrid neurosymbolic systems, where the primary computation is symbolic, and the neural network serves to parameterize the symbolic space. We take a different approach, one where symbolic operations happen in vector space. In our system, neural and symbolic computations are **unified** into a single space; we multiply and add vector-embedded symbolic structures instead of multiplying and adding individual neurons.

We introduce a new technique for representing trees in vector space called Sparse Coordinate Trees (SCT). SCT allows us to perform structural operations: transformations which change the structure of an object without changing the content. This is a crucial aspect of compositionality, where the structure and content can be transformed independently. We extend the Differentiable Tree Machine (DTM), a system which operates over binary trees in vector space, into the Sparse Differentiable Tree Machine (sDTM) to improve performance and applicability to a larger variety of tasks[2]. While DTM processes vector-embedded binary trees as the primitive unit of computation, the order of operations and argument selection is governed by a Transformer. We present results showing that this unified approach retains many of the desirable properties of more brittle symbolic models with regards to generalization, while remaining flexible enough to work across a far wider set of tasks. While fully neural architectures or hybrid neurosymbolic techniques excel at certain types of generalization, we find that DTM, with its unified approach, excels across the widest array of shifts.

The main contributions from this paper are:

- Sparse Coordinate Trees (SCT), a method for representing binary trees in vector space as sparse tensors. (§3.1)
- Bit-Shift Operating — systematic and parallelized tree operations for SCT. (§3.2)
- The introduction of Sparse Differentiable Tree Machine (sDTM), architectural improvements to the DTM to leverage SCT and drastically reduce parameter and memory usage. (§4)
- Techniques to apply DTM to seq2seq tasks by converting sequences into trees. (§4.5)
- Empirical comparisons between sDTM and various baselines showing sDTM's strong generalization across a wide variety of tasks. (§5)

## 2 Related Work

Work leveraging the generalizability of tree structures has a long history across Computer Science, Linguistics, and Cognitive Science [9, 45, 57, 64, 68]. Much of classical NLP aims to extract structured representations from text like constituency or dependency parses [for overview: 13, 42]. More recent work has shown the representations learned by sequence-to-sequence models without structural supervision can recover constituency, dependency, and part of speech information from latent representations in machine translation and language models [3, 6]. While those analyses show

---

[2]Code available at https://github.com/psoulos/sdtm.

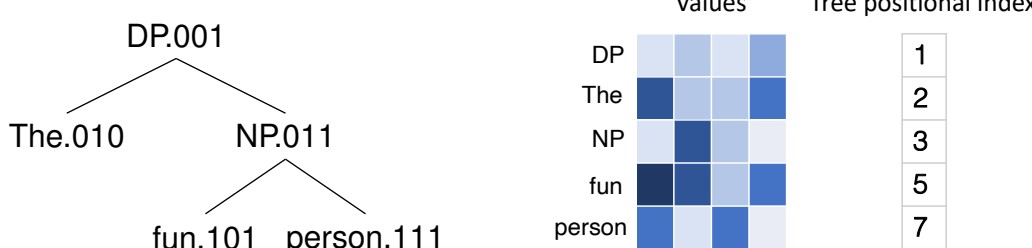

Figure 2: An example representation using Sparse Coordinate Trees (SCT). The values are N-dimensional vectors, and the tree positional indices are integer representations of positions in the tree. The absent child nodes of "The" (indices 4 and 6) are skipped with SCT.

structural information is encoded, they stop short of showing that the representations themselves are tree-structured. Analyses inspired by Tensor Product Representations [46, 66] and chart parsing [48] give an account of how representations become somewhat tree-structured over the course of training.

Despite the apparent emergence of semi-structured representations in Transformers and LSTMs, these architectures still appear to struggle with the kinds of structural generalization that come easily to humans [36, 31, 29]. A variety of approaches try to tackle this problem through meta-learning [37, 11], data augmentation [1], or decomposing the task into separate parts [56, 41]. The Relative-Universal Transformer [12] combines relative positional embeddings with a recurrent component, in an effort to emphasize local structures while allowing for unbounded computation.

Explicitly tree structured network architectures have been introduced for RNNs [65], LSTMs [70, 16], and Transformers [72, 61]. However, these variants often do not outperform their unstructured counterpart on out-of-distribution challenges [67]. This may be because generalization requires both structured representations and operations that respect that structure. A separate line of work considers neural architectures that are used to parameterize components of a symbolic system [33, 8] or fuzzy/probabilistic logic [75, 2, 73, 14]. Similar to how vectors are embedded in trees in our work, some work embeds vectors within logical systems [54, 43]. Logic approaches to structure learning are also an active area of research [47, 59]. Other approaches leverage explicit stack operations [17, 24, 27, 76]. NQG from Shaw et al. [58] combines the outputs from neural and symbolic models by inducing a grammar, but deferring to T5 [53] when that grammar fails. However the grammar's induction method has polynomial complexity with both dataset size and sequence length, which limits its application to larger tasks.

Vector Symbolic Architectures (VSAs) implement symbolic algorithms while leveraging high dimensional spaces [52, 21, 28, 35]. VSAs are similar to uniform neurosymbolic approaches, although VSAs commonly lack a learning component. Our work extends that of Soulos et al. [67] which can be viewed as integrating Deep Learning and VSAs. They introduce the Differentiable Tree Machine for Tree-to-Tree transduction. Here we instantiate a sparse Sequence-to-Sequence version with far fewer parameters and improved memory efficiency.

## 3   Differentiable Tree Operations Over Sparse Coordinate Trees

Representing trees in vector space enables us to perform differentiable structural operations on them. Soulos et al. [67] used Tensor Product Representations (TPRs) [64] for this purpose. TPRs use the tensor (or outer) product to represent trees in vector space (§A.1). Use of an outer product leads to a representation dimensionality that is multiplicative with both the embedding dimensionality and the number of possible tree nodes. Additionally, the number of nodes is itself an exponential function of the supported depth. This makes TPRs difficult to use in practice, given available memory is quickly exceeded as tree depth increases.

In this section, we introduce Sparse Coordinate Trees (SCT), a new schema for representing trees in vector space. We then define a library of parallelized tree operations and how to perform these operations on SCT.

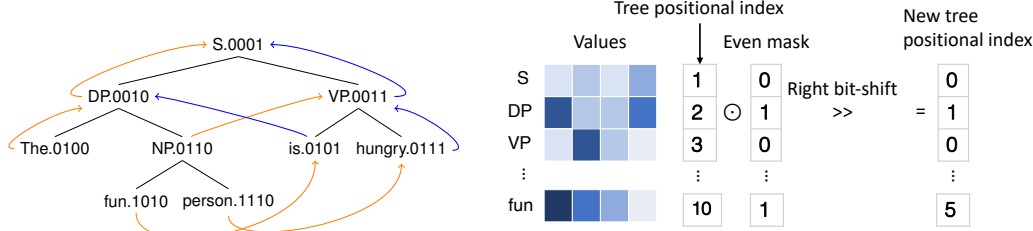

Figure 3: Left: Performing `left` (orange) and `right` (blue). Right: visualizing the `left` transformation which results in DP being placed at the root. Tree positional indices of $0$ and their corresponding values are discarded.

## 3.1 Sparse Coordinate Trees (SCT)

Like TPRs, we want an encoding for trees that factorizes the representation into subspaces for *structure* and *content* respectively. This approach to representational spaces differs from models like an RNN and Transformer, which represent structure and content jointly in an unfactorized manner. By separating the structure and content subspaces a priori, we can operate over these two spaces independently. This decision is motivated by the fact that distinct treatment of these spaces is an essential aspect of compositionality.

We derive our tree representation scheme from the sparse coordinate list (COO) format. COO stores tensor data in tuples of (indices [integers], values [any format], size [integers]). The indices are N-dimensional to simulate a tensor of arbitrary shape (e.g. including dimensions such as batch or length). When an index is not indicated in indices, it is assumed that the corresponding value is $0$.

We give structural meaning to COO representations by defining one dimension of indices as the tree position occupied by a value vector. Our tree addressing scheme is based on Gorn addresses [23]: to get the tree position from an index, convert the index to binary and read from right to left. A left-branch is indicated by a $0$ and a right branch by a $1$. To distinguish between leading 0s and left-branches (e.g. 010 vs 10), we start our addressing scheme at $1$ instead of $0$. This indicates that all 0s to the left of the most-significant 1 are unfilled and not left-branches. Figure 2 shows an example encoding of a tree with this approach. SCT can be viewed as a TPR with certain constraints, and Section A.1 defines this equivalence and formally describes the memory savings.

Section 5.2 discusses the performance, memory, and parameter comparison between DTM models which use standard TPRs and SCT.

## 3.2 Differentiable Tree Operations

To operate on the trees defined in the previous section, we need a set of functions. We use a small library of only three: left-child (`left`), right-child (`right`), and **cons**truct (`cons`) a new tree from a left and right subtree.[3] Although these three functions are simple, along with the control operations of conditional branching and equality-checking, these five functions are Turing complete [45].

In addition to saving memory, SCT also provides a more efficient method for performing differentiable tree operations. The operations defined in Soulos et al. [67] require precomputing, storing, and applying linear transformations for `left`, `right`, and `cons`. Since our values and tree positional indices are kept separate, we can compute the results of `left`, `right`, and `cons` dramatically more efficiently using indexing, bit-shifts, and addition.

Figure 3 shows how we can perform `left` directly on SCT. `left` is performed by indexing the even indices (i.e. those with a 0 in the least significant bit, which targets all of the nodes left of the root) and their corresponding values, then performing a right bit-shift on the indices. `right` is symmetrical, except that we index for the odd positional indices and ignore position 1 in order to remove the previous root node. `cons` is performed by left bit-shifting the positional indices from the left- and

---

[3]In LISP and expert systems literature, `left` is referred to as `car`, and `right` is referred to as `cdr`.

right-subtree arguments, then adding 1 to the newly shifted indices for the right argument. A new value $s$ can be provided for the root node.

Our network also needs to learn a *program* over multiple operations differentiably. This involves the aforementioned structured operations, as well as differentiable selection of which operation to perform and on which trees. We take weighted sums over the three operations `left`, `right`, and `cons`, as well as over potential trees. Specific details are discussed in the next section. The result of our weighted sum is coalesced, which removes duplicate positional indices by summing together all of the values that share a specific index. Formally, define the trees over which to perform `left` $T_L$, `right` $T_R$, and `cons` $T_{CL}$ & $T_{CR}$; $\vec{T} = [T_L; T_R; T_{CL}; T_{CR}]$. We also take a new value $s \in \mathbb{R}^d$ to be inserted ($\otimes$) at the new root node of the `cons` operation, and a vector of operation weights $\vec{w} = (w_L, w_R, w_C)$ which sum to 1.

$$O(\vec{w}, \vec{T}, s) = w_L \texttt{left}(T_L) + w_R \texttt{right}(T_R) + w_C(\texttt{cons}(T_{CL}, T_{CR}) + s \otimes r_1) \tag{1}$$

## 4 The Sparse Differentiable Tree Machine (sDTM)

Our work extends the Differentiable Tree Machine (DTM) introduced in Soulos et al. [67] with the Sparse Differentiable Tree Machine (sDTM). While similar to the original at a computational level, sDTM represents a different implementation of these concepts that make it dramatically more parameter and memory efficient. We also introduce techniques to apply sDTM to tasks with sequence input and output (seq2seq).

### 4.1 Overview

sDTM uses our Sparse Coordinate Trees schema across its components. Like the original DTM, our model is comprised of an agent, interpreter, and memory (illustrated in Figure 4). The Interpreter performs Equation 1 by applying the bit-shifting tree operations from Section 3.2 and weighting the result. The output from the interpreter is written to the next available memory slot, and the last memory slot is taken as the output.

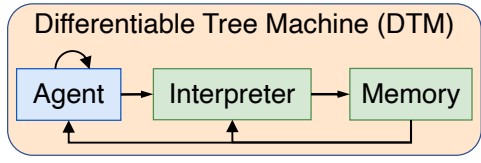

Figure 4: A schematic of how the three core components of the DTM (agent, interpreter, and memory) relate to each other. Adapted from Soulos et al. [67].

The Agent is a Transformer encoder that takes an encoding of the memory as input and produces the inputs for Equation 1: $\vec{w}$, $\vec{T}$, and $s$. Two special tokens, <OP> and <ROOT>, are fed into the Agent to represent $\vec{w}$ and $s$. Each time a tree is written to memory, a fixed-dimensional encoding of that tree is produced and fed as a new token to the agent (§4.2). The agent soft-selects tree arguments for the interpreter, $\vec{T}$, by performing a weighted sum over the trees in memory. Figure 6 in the Appendix contains a detailed diagram showing the flow of information for one layer of sDTM.

The agent which implicitly parameterizes the conditional branching and control flow of the program is modeled by a Transformer, and it is possible for sDTM to face some of the generalization pitfalls that plague Transformers. The design of sDTM encourages compositional generalization through differentiable programs, but it does not strictly enforce the constraints of classical symbolic programs. As the results in Section 5 show, sDTM can learn generalizable solutions to some tasks despite the presence of a Transformer, but on some other tasks the issues with generalization are still present.

### 4.2 Pooling by attention

Each tree in memory needs to have a fixed-dimensional encoding to feed into the agent regardless of how many nodes are filled. Commonly this is done via pooling, like taking the means of the elements in the tree, or a linear transformation in the case of the original DTM. Instead, we use Pooling by Multi-headed Attention (PMA) [38], which performs a weighted sum over the elements, where the weight is derived based on query-key attention.

Attention is permutation invariant to the ordering of key and value vectors, but it is important that our pooling considers tree position information. To enforce this, we convert the position indices to their binary vector representation $\vec{b}$. This leads to an asymmetrical vector with only positive values, so instead we represent left branches as $-1$ and keep right branches as $+1$. For example, position $5 \rightarrow [0, 0, 0, 0, 0, 1, 0, 1] \rightarrow [0, 0, 0, 0, 0, 1, -1, 1]$. The input to our pooling function is the concatenation of this positional encoding $\vec{b}$ with the token embedding $\vec{x}$ at that position: $[\vec{x}; \vec{b}]$. This method for integrating token and node position is similar to tree positional encoding from Shiv and Quirk [60], except that we use concatenation and a linear transformation to mix the content and position information instead of addition.

Unlike standard self attention, we use a separate learnable parameter for our query vector $\vec{q} \in \mathbb{R}^{\text{num\_heads} \times \text{key\_dim}}$. We pass $[\vec{x}; \vec{b}]$ through linear transformations to generate keys $\vec{k} \in \mathbb{R}^{\text{num\_heads} \times \text{key\_dim}}$ and values $\vec{v} \in \mathbb{R}^{\text{num\_heads} \times \text{value\_dim}}$. The result of this computation is always $z \in \mathbb{R}^{\text{num\_heads} \times \text{value\_dim}}$ given that $\vec{q}$ is fixed and does not depend on the input. The rest of the computation is identical to a Transformer with pre-layer normalization [74].

### 4.3 Tree Pruning

While Sparse Coordinate Trees mean that trees with fewer filled nodes take up less memory, the way our model blends operations results in trees becoming dense. The interpreter returns a blend of all three operations at each step, including the `cons` operation which increases the size of the representation by combining two trees. In practice even as the entropy of the blending distribution drops, the probability of any operation never becomes fully 0. This means that over many steps, trees start to become dense due to repeated use of `cons`. In order to keep our trees sparse, we use pruning: only keeping the top-$k$ nodes as measured by magnitude. $k$ is a hyper-parameter that can be set along with the batch size depending on available memory.

### 4.4 Lexical Regularization

To aid lexical generalization, we add noise to our token embeddings. Before feeding an embedded batch into the model, we sample from a multi-variate standard normal for each position in each tree, adding the noise to the embeddings as a form of regularization [5]. Ablation results showing the importance of this regularization are available in Appendix A.3.

### 4.5 Handling Sequential Inputs and Outputs

**seq2tree** The original `DTM` can only be applied to tasks where a tree structure is known for both inputs and outputs. Here we provide an extension to allow `DTM` to process sequence inputs. To do this we treat each input token as a tree with only the root node occupied by the token embedding. We then initialize the tree memory with $N$ trees, one for each token in the input sequence. Figure 5 left depicts the initial memory state for a sequence. The agent's attention mechanism is permutation-invariant, so in order to distinguish between two sequences which contain the same tokens but in different orders, we apply random sinusoidal positional encodings to the first $N$ tokens passed to the agent [40, 55]. Random positional encodings sample a set of increasing integers from left-to-right instead of assigning a fixed position to each token. The purpose of `left` and `right` is to extract subtrees. Since in our seq2tree setting the input sequence is processed in a completely bottom-up manner, we restrict the agent and interpreter to only have a single operation: `cons`. Use of a single operation to construct new trees from subtrees aligns the `DTM` theoretically with the Minimalist Program [10], which addresses natural language's compositionality in terms of a single operation: merge.

**seq2seq** To handle sequence inputs and outputs we convert the output sequence to a tree. One method to convert the output sequence into a tree is to use a parser. Alternatively, when a parser is not available, we can embed a sequence as the **l**eft-**a**ligned leaves at **u**niform **d**epth (LAUD). Figure 5 right shows how an output sequence can be embedded using LAUD. Since all of the non-terminal nodes are the same, we can hardcode the root argument to `cons`. We insert a special token <EOB> to signify the end of a branch, similar to an <EOS> token.

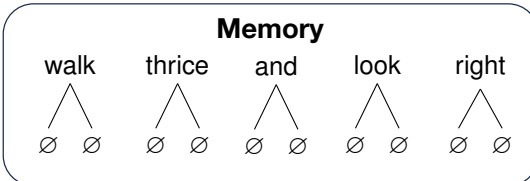
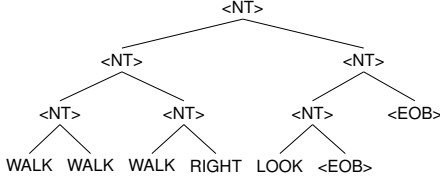

Figure 5: Left: The memory state is initialized as a sequence of trees where only the root node contains a token. Right: An output sequence is embedded in a tree using the left-aligned uniform-depth (LAUD) scheme. <NT> and <EOB> are special tokens not in the original output sequence.

## 5 Results

### 5.1 Baselines

We consider models that are trained from scratch on the datasets they're evaluated on; while the compositional capabilities of large pre-trained models are under active debate [32], we are interested in the compositional abilities of the underlying architecture — rather than those that may result from a pre-training objective. We compare sDTM to two fully neural models, a standard Transformer [71] as well as a relative universal Transformer (RU-Transformer) which was previously shown to improve systematic generalization on a variety of tasks [12]. We also compare our model to a hybrid neurosymbolic system, NQG [58, 69], a model which uses a neural network to learn a quasi-synchronous context-free grammar [62]. NQG was introduced alongside NQG-T5, which is a modular system that uses NQG when the grammar produces an answer and falls back to a fine-tuned large language model T5 [53]. As mentioned at the beginning of this section, we only compare to NQG in this paper since we want to evaluate models that have not undergone significant pre-training.[4] Details related to data preprocessing (§A.4), model training (§A.6, §A.8), compute resources (§A.9), and dataset details (§A.10) are available in the Appendix. For all datasets, the reported results are the best exact match accuracies on the test set over five random seeds. Additional data on means and standard deviations across the five runs are shown in Section A.7.

For each task, we test whether models generalize to samples drawn from various data distributions. Independent and identically distributed (**IID**) samples are drawn from a distribution shared with training data. We evaluate several out-of-distribution (OOD) shifts. **One-shot** lexical samples, while drawn from the same distribution as the training data, contain a word that was only seen in a single training sample. Similarly, **Zero-shot** lexical samples are those where the model is not exposed to a word at all during training. **Structural/length** generalization tests whether models can generalize to longer sequences (length) or nodes not encountered during training (structural). **Template** generalization withholds an abstract n-gram sequence during training, and then each test sample fits the template. Finally, maximum compound divergence (**MCD**) generates train and test sets with identical uni-gram distributions but maximally divergent n-grams frequencies [29]. Although models are often tested on a single type of generalization, we believe evaluating a model across a broad array of distributional shifts is essential for characterizing the robustness of its generalization performance.

### 5.2 Performance Regression (Active↔Logical)

Active↔Logical is a tree2tree task containing input and output trees in active voice and logical forms [67]. Transforming a tree in active voice to its logical form simulates semantic parsing, and transforming a logical form tree to active voice simulates natural language generation. For this dataset, there are three test sets: IID, 0-shot lexical, and structural. In addition to the baselines listed in the previous section, we also compare our modified sDTM to the original DTM. This enables us to confirm that our proposed changes to decrease parameter count and memory usage while increasing inference speed does not lead to a performance regression. The results are show in Table 1.

---

[4]NQG uses pre-trained BERT embeddings [15]; it is unknown how much this pre-training helps the method.

Table 1: **Active↔Logical accuracy.** Results are the best performance over five runs. The test sets are divided into IID, and OOD sets 0-shot lexical and structural. Parameter and memory usage is shown for the original DTM with TPRs and our proposed sparse DTM with and without pruning. Our modifications reduce the parameter count by almost two orders of magnitude. *NQG was trained on a seq2seq version without parantheses because it was not able to learn the tree2tree training set.

| Model | Split | | | | | |
|---|---|---|---|---|---|---|
| | IID | 0-Shot Lexical | Structural | | | |
| Transformer | 1.0 | 0.0 | 0.0 | | | |
| RU-Transformer | 1.0 | 0.0 | .12 | | | |
| NQG* | .45 | 0.0 | 0.0 | | | |
| | | | | Parameters | Memory (GB) | Relative Speed |
| Original DTM | 1.0 | 1.0 | 1.0 | 72M | 12.3 | 34 |
| sDTM | 1.0 | 1.0 | 1.0 | 1M | 9.7 | 2.5 |
| sDTM (pruned k=1024) | 1.0 | 1.0 | .95 | 1M | 1.8 | 1 |

The various DTM models and Transformers all perform perfectly on the IID test set. NQG struggles to learn the Active↔Logical task, an example of the brittleness of hybrid neurosymbolic systems. Only the DTM variants succeed on the OOD test sets. As anticipated, the RU-Transformer performs better than the standard Transformer with regards to structural generalization.

Comparing the original DTM to sDTM without pruning, we see a 70x reduction in parameter count from pooling by attention, a 20% reduction in memory usage from fewer parameters and SCT, as well as a roughly 13x speedup. We are able to gain even further memory savings and speed improvements due to the pruning method. The final two rows show that the pruning method has no impact on lexical generalization and a minor impact on structural generalization, while reducing memory usage by 5x and improving speed by 2.5x. The results from this experiment confirm that sDTM is capable of matching DTM performance on a previous baseline. However, since both DTM and sDTM perform near ceiling, it is difficult to isolate the effect of the proposed changes in this paper. We will investigate this question further in Section 5.5. Next, we turn to tasks where the original DTM could not be used.

## 5.3 Scalability (FOR2LAM)

FOR2LAM is a tree2tree program translation task to translate an abstract syntax tree (AST) in an imperative language (FOR) to an AST in a functional language (LAM) [7]. Due to the depth of the trees in this dataset, DTM is unable to fit a batch size of 1 into memory. This makes FOR2LAM a good dataset to test the scalability of sDTM to more complex samples. We augment the FOR2LAM dataset with a 0-shot lexical test set. During training, only two variable names appear: 'x' and 'y'. For the 0-shot test, we replace all occurrences of x in the test set with a new token 'z'. We are unable to test DTM on FOR2LAM because a batch size of 1 does not fit into memory due to the depth of the trees in the dataset.

Results on FOR2LAM are shown on the left side of Table 2. NQG suffers with scale (see A.8), and we were unable to include results for it on FOR2LAM due to training and evaluation exceeding 7 days. All other models do well on the in-distribution test set, but only DTM is able to achieve substantive accuracy on the 0-shot lexical test. DTM's performance is impressive given work on data augmentation has shown the difficulty of few-shot generalization is inversely proportional to vocabulary size [50], with smaller vocabulary tasks being more challenging. This 0-shot challenge is from 2 variables (x, y) to 3 (x, y, z), making it difficult enough that both transformer variants score 3%.

## 5.4 Seq2Tree (GeoQuery)

GeoQuery is a natural language to SQL dataset [77] where a model needs to map a question stated in natural language to a correctly formatted SQL query, including parentheses to mark functions

Table 2: **Accuracies on FOR2LAM and GeoQuery.** Results are the best performance over five runs. NQG cannot be evaluated on FOR2LAM because it takes over a week to train. [†]Results taken from Shaw et al. [58]. [*]We report the results from a replication study of NQG where the result on the Length split differed substantially from the original result [69].

| Model | FOR2LAM | | GeoQuery | | | |
| | IID | 0-shot lexical | IID | Length | Template | TMCD |
| --- | --- | --- | --- | --- | --- | --- |
| Transformer | 1.0 | .03 | .88 | .26 | .79 | .40 |
| RU-Transformer | 1.0 | .03 | .87 | .25 | .77 | .37 |
| NQG[†] | – | – | .76 | .37/.26[*] | .62 | .41 |
| sDTM | 1.0 | .61 | .73 | .20 | .20 | .36 |

and arguments. We use the parentheses and function argument relationship as the tree structure for our output. In this format, GeoQuery is a seq2tree task, and we follow the description from Section 4.5. We use the same preprocessing and data as Shaw et al. [58]. The TMCD split for GeoQuery [58] extends MCD to natural language datasets instead of synthetic languages. GeoQuery is a very small dataset, with a training set containing between 440 and 600 samples, depending on the split. Like FOR2LAM, we are unable to test DTM on GeoQuery because a batch size of 1 does not fit into memory due to the depth of the trees in the dataset.

Results for GeoQuery are shown on the right side of Table 2. This is the most difficult task that we test because of the small training set, and the natural language input is not draw from a synthetic grammar. Given this, a potential symbolic solution to this task might be quite complex. We find that both NQG and DTM perform worse than the two Transformer variants on the IID test set. This also holds true for the Template split, where Transformers outperform the neurosymbolic models. On the Length and TMCD splits, all of the baselines achieve roughly the same performance while DTM performs slightly worse — the degree of variation in the input space and small training set appear to make it difficult for sDTM to find a compositional solution.

It is worth noting that there is substantial room for improvement across every model on GeoQuery. The small dataset with high variation poses a problem for both compositional methods of sDTM and NQG. It is possible that with sufficient data, GeoQuery's latent compositional structure could be identified by NQG and DTM, but the released GeoQuery dataset has only on the order of 500 training examples. Given all methods struggle to model the IID split, we refrain from drawing substantive conclusions based on minor differences in accuracy on this single task in isolation from the rest of the results.

## 5.5 Seq2Seq (SCAN)

SCAN is a synthetic seq2seq task with training and test variations to examine out-of-distribution generalization [36]. To process seq2seq samples, we follow the description in Section 4.5. We compare two methods for embedding the output sequence into a tree by writing a parser for SCAN's output and comparing this to the left-aligned uniform-depth trees (LAUD). In addition to the standard test splits from SCAN, we introduce a 0-shot lexical test set as well.

Since the trees in SCAN are not very deep, we are able to compare sDTM to DTM to isolate the effect of pooling by attention (§4.2). We modify the original DTM to handle sequential inputs and outputs as described in Section 4.5. Replacing the linear transformation in DTM with pooling by attention in sDTM leads to drastically better results; DTM is unable to perform well even on the simple IID split, whereas sDTM performs well across many of the splits.

All baselines perform well on the IID test set, showing that they have learned the training distribution well. Transformer variants perform poorly on lexical, length, and MCD splits. The Transformers and sDTM perform well on the Template split while NQG completely fails. Along with the results from GeoQuery, which showed weak sDTM performance on the Template split and strong performance from both Transformers, it seems that the Transformer architecture is robust under template shifts between training and testing. sDTM is the only model to perform well on the 0-shot lexical test set, whereas NQG is the only model able to perform well on the MCD test set. The two sDTM rows

Table 3: **SCAN accuracy.** Results are the best performance over five runs. MCD scores are calculated as the average of the three MCD splits. [†]Results from Shaw et al. [58]. [*]Results from Sun et al. [69].

| Model | Split | | | | | |
|---|---|---|---|---|---|---|
| | IID | 1-shot lexical | 0-shot lexical | Length | Template | MCD |
| Transformer | 1.0 | .08 | 0.0 | .07 | 1.0 | .02 |
| RU-Transformer | 1.0 | .11 | 0.0 | .19 | 1.0 | .01 |
| NQG[†] | 1.0* | 1.0 | 0.0 | 1.0 | 0.0* | 1.0 |
| sDTM (parse trees) | 1.0 | .99 | .99 | .75 | .95 | .03 |
| sDTM (LAUD trees) | 1.0 | .87 | .98 | .06 | .98 | 0.0 |
| DTM (parse trees) | 0.0 | 0.0 | 0.0 | 0.0 | 0.0 | 0.0 |

compare models trained with output trees from a parser or LAUD encoding. The main performance difference is on the Length split, where the structurally relevant information in the parse trees is necessary for sDTM to perform well. It is not necessary to have structured input for the model to perform well on length generalization as long as the output is structured.

## 6    Conclusions

We introduced the Sparse Differentiable Tree Machine (sDTM) and a novel schema for efficiently representing trees in vector space: Sparse Coordinate Trees (SCT). Unlike the fully neural and hybrid neurosymbolic baselines presented here, sDTM takes a unified approach whereby symbolic operations occur in vector space. While not perfect — sDTM struggles with MCD and Template shifts, as well as the extremely small GeoQuery dataset — the model generalizes robustly across the *widest variety* of distributional shifts. sDTM is also uniquely capable of zero-shot lexical generalization, likely enabled by its factorization of content and structure.

While these capacities for generalization are shared with the original DTM, our instantiation is computationally efficient (representing a 75x reduction in parameters) and can be applied to seq2seq, seq2tree, and tree2tree tasks. Our work reaffirms the ability of neurosymbolic approaches to bridge the flexibility of connectionist models with the generalization of symbolic systems. We believe continued focus on efficient neurosymbolic implementations can lead to architectures with the kinds of robust generalization, scalability, and flexibility characteristic of human intelligence.

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

# A  Appendix

## A.1  Sparse Coordinate Trees as Tensor Product Representations

This section shows that Sparse Coordinate Trees is the same as a TPR with the constraint that the role basis is the standard basis. TPRs define structural positions as role vectors $r_i \in \mathbb{R}^{d_r}$, and the content that fills these positions is defined by filler vectors $f_i \in \mathbb{R}^{d_f}$. For a particular role and filler pair, the filler $f_i$ is *bound* to the role $r_i$ using the tensor/outer product: $f_i \otimes r_i \in \mathbb{R}^{d_f \times d_r}$. The representation of an entire structure is the sum over all $N$ individual filler-role pairs: $T = \sum_{i=1}^{N} f_i \otimes r_i \in \mathbb{R}^{d_f \times d_r}$. As shown in the previous two equations, the dimensionality of a single filler-role pair is equal to the dimensionality of an entire structure: both have dimensionality $\mathbb{R}^{d_f \times d_r}$. This means that a tree with only a filled root node takes up the same memory as a dense tree with every node filled. An important requirement for TPRs is that the role vectors must be linearly independent; this ensures that a filler can be *unbound* from a role without introducing noise using the inner product: $f_j = T r_j^+$, where $\{r_i^+\}_i$ is the basis dual to $\{r_i\}_i$. Previous work typically used randomly initialized and frozen orthonormal vectors to define the role basis. By defining our role vectors in a sparse manner as opposed to random initialization, we can greatly reduce the memory used by TPRs.

Classic symbolic data structures grow in memory linearly with the number of filled positions. It is possible to replicate this behavior with TPRs by defining the role vectors to be the standard one-hot basis, which is orthonormal by definition. The $i$-th element of role vector $r_i$ is 1, and the other elements are 0. When a filler and role vector are both dense, the resulting bound vector is also dense. When the role vector is one-hot, the resulting bound vector is 0 everywhere except for column $i$ which corresponds to the value 1 in $r_i$. By using a sparse tensor representation that only keeps track of dimensions that are not equal to 0, we can reduce the memory usage of TPRs to linear growth that scales with the number of filled positions, like a classical symbolic data structure. This however forgoes a motivating desideratum for the design of TPRs, that roles (and not just fillers) have similarity relations that support generalization across structural positions.

We can additionally improve the efficiency by refraining from performing the outer product. Since we are not performing a **tensor** product, this technique is only implicitly a **Tensor** Product Representation. Instead, we can keep the filler and role vectors in two aligned lists. A filler is bound to a role by sharing an index in our aligned lists. This is equivalent to the *binding* and *unbinding* from classical dense TPRs without having to perform multiplication.

Since we are not performing an outer product, instead of storing sparse role vectors, we can simply store a role integer, where the integer corresponds to the one-hot dimension. We derive a tree addressing scheme based on Gorn addresses [23]. In our scheme, addresses are read from right to left, giving the path from the root where a left-branch is indicated by a 0 and a right-branch is indicated by a 1. We need a way to distinguish between leading 0s and left-branches (e.g., 010 vs. 10), so we start our addressing scheme at 1 instead of 0. This indicates that all 0s to the left of the left-most 1 are unfilled and not left-branches; the left-most 1 and all preceding 0s are ignored when decoding the path-from-root. Figure 2 shows an example encoding of a tree in the sparse implicit approach.

We can compare the memory requirements of the Sparse Coordinate Tree encoding used in the sDTM to the memory requirements of the full TPRs used in the original DTM of Soulos et al. [67]. A TPR uses the same amount of memory regardless of the number of filled nodes. As with all sparse tensor formats, the memory savings arise when there are many zeros. In a dense tree where every node is occupied, the classical dense TPR approach is actually more efficient: the SCT's value list has the same total dimension as the classical TPR, but, in addition, the SCT encoding includes the list of filled-node addresses.

## A.2  Agent Figure

See Figure 6.

## A.3  Lexical Regularization Ablation

To see the importance of adding noise to our input embeddings as defined in Section 4.4, we show the performance of sDTM with and without this regularization in Table 4.

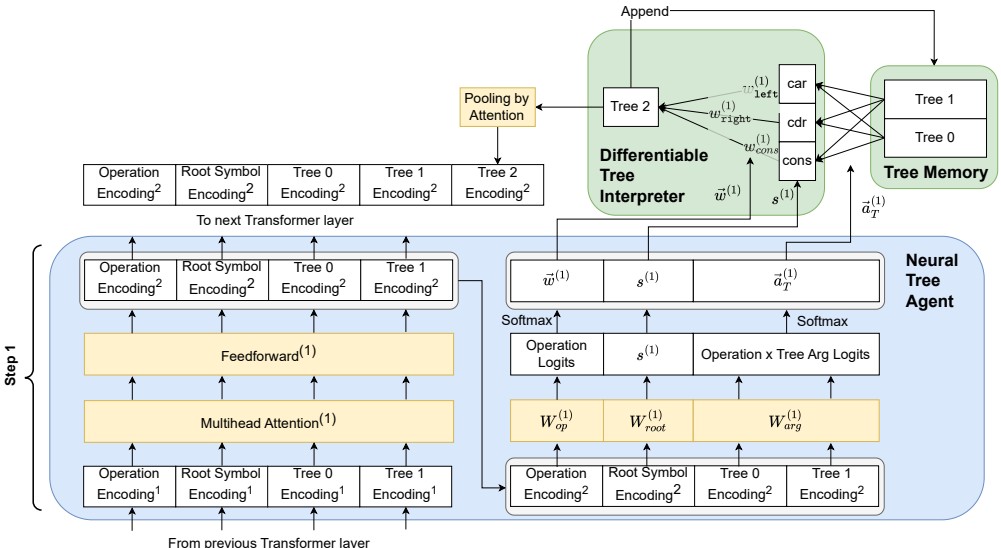

Figure 6: Adapted from Soulos et al. [67]. One step of DTM is expanded to show how the agent produces the input to the interpreter. The interpreter then writes the output to memory and encodes the output for the agent. Parts of the architecture with learnable parameters are indicated in yellow. The agent uses three linear transformations on top of a standard Transformer encoder layer to parameterize the inputs to the interpreter. The superscript indicates the layer number and refers to parameters and activations that are exclusive to this layer.

Table 4: Comparing sDTM's accuracy on SCAN 1-shot lexical OOD generalization with and without lexical regularization. We use LAUD to embed the output sequence in a tree.

| | |
|---|---|
| With noise | .87 |
| Without noise | 0.0 |

## A.4 Dataset Preprocessing

We preprocessed GeoQuery according to the steps from Shaw et al. [58]. FOR2LAM and GeoQuery both contain non-binary trees, which we convert to binary form using Chomsky normal form. When a new node is inserted to make a branch binary, we use the token <NT>. For output sequences with length one embedded according to left-aligned uniform-depth, we make the single token the left child of a new <NT> root node.

## A.5 0-shot Lexical Test Generation

For both FOR2LAM and SCAN, we introduce 0-shot lexical tests. For FOR2LAM, we do this by replacing every occurrence of 'x' in the test set with a new token 'z'. For the SCAN 0-shot set, we start with the 1-shot lexical test set and remove the sample containing the 1-shot word 'jump'. We alter the output vocabulary to use the same tokens as the input vocabulary, since it is impossible for a word level model to translate between an input and output word without any exposure to that word.

## A.6 DTM Training Details

When applicable, we adopt the hyperparameters from Soulos et al. [67]. Below we list the newly introduced hyperparameters and changes we made to existing parameters.

Soulos et al. [67] set the dimensionality of the embeddings to be equal to the size of vocabulary. This works for the datasets with small vocabulary examined in the original paper. We keep this setting for

Table 5: **Summary statistics for Active↔Logical.** Mean and standard deviation accuracies are shown.

| Model | Split | | |
|---|---|---|---|
| | IID | 0-Shot Lexical | Structural |
| Original DTM | $1.0 \pm 0.0$ | $1.0 \pm 0.0$ | $1.0 \pm 0.0$ |
| sDTM | $1.0 \pm 0.0$ | $1.0 \pm 0.0$ | $1.0 \pm 0.0$ |
| sDTM (pruned k=1024) | $1.0 \pm 0.0$ | $1.0 \pm 0.0$ | $.86 \pm .04$ |

Table 6: **Summary statistics for FOR2LAM and GeoQuery.** Mean and standard deviation accuracies are shown.

| | **FOR2LAM** | | **GeoQuery** | | | |
|---|---|---|---|---|---|---|
| Model | IID | 0-shot lexical | IID | Length | Template | TMCD |
| sDTM | $1.0 \pm 0.0$ | $.43 \pm .11$ | $.68 \pm .01$ | $.19 \pm .01$ | $.16 \pm .03$ | $.35 \pm 0.0$ |

Active↔Logical, but set the embedding dimension to 64 for FOR2LAM, and 128 for GeoQuery and SCAN. We also changed the loss function from mean-squared error to cross entropy.

For each new task, we need to decide how many layers to use for sDTM. We followed the heuristic of doubling the max tree depth for the models with sequence input and quadrupling the number of layers for tree input. This leads to 56 layers for FOR2LAM, 22 layers for GeoQuery, and 14 layers for SCAN.

Pooling by multi-headed attention 4.2 introduces new hyperparameters such as number of pooling heads and pooling key dimensionality, and we set the value of these to be the same as the Transformer hyperparameters for the agent. Tree pruning 4.3 introduces a new hyperparameter $k$ for the maximum number of nodes to keep. In general, a larger $k$ is better but uses more memory. For Active↔Logical we set $k = 1024$, for FOR2LAM $k = 1024$, for GeoQuery $k = 2048$, and for SCAN $k = 256$. With the memory savings from SCT, pooling by multi-headed attention, and pruning, we increase the batch size from 16 to 64. We also increased the agent's model dimension to 256 with 8 heads of attention due to the memory savings except for Active↔Logical where we matched the original hyperparameters.

Random positional embeddings (RPE) also introduce a new hyperparameter for the max input integer, and we set this to be double the max input length. This leads to an RPE hyperparameter of 44 for GeoQuery and 18 for SCAN.

We noticed that randomly initializing and freezing our embedding vectors was essential for sDTM to achieve 0-shot generalization on SCAN.

For the results, we reported the best run of 5 random seeds. Like DTM, sDTM suffers from high variance. Some runs get stuck in local optima and fail to achieve moderate performance on the training set, which leads to poor performance on the test sets. This is a known issue with models that use superposition data structures, and reporting the best run over a number of random seeds has been previously used [76, 17].

### A.7 DTM **Summary Statistics**

Mean and standard deviation accuracies are shown in Tables 5, 6, and 7.

### A.8 **Baseline Training Details**

**NQG:** Active↔Logical rule induction used the following hyperparameters: sample size=training set size, terminal code length=8, allow repeated nts=True. The terminal code length setting was

Table 7: **Summary statistics for SCAN.** Mean and standard deviation accuracies are shown.

| Model | Split | | | | | |
|---|---|---|---|---|---|---|
| | IID | 1-Shot Lexical | 0-Shot Lexical | Length | Template | MCD |
| sDTM (parse trees) | .80±.39 | .51±.39 | .80±.34 | .34±.30 | .19±.38 | .02±.01 |
| sDTM (LAUD trees) | 1.0±0.0 | .70±.15 | .75±.22 | .04±.01 | .84±.12 | 0.0±0.0 |
| DTM (parse trees) | 0.0±0.0 | 0.0±0.0 | 0.0±0.0 | 0.0±0.0 | 0.0±0.0 | 0.0±0.0 |

obtained via grid search over the values 1, 8, 32. For the actual training of the model we follow the hyperparameters utilised by [69, 58]. FOR2LAM used the same hyperparameters with the exception of sample size which had to be set to 1000 as additional increases became computationally intractable. Even under these settings rule induction took 42 hours on a machine with 64gb of ram. Writing the training set would take an additional week of processing time, which we considered computationally too expensive.

**Transformer:** We followed the same hyperparameters obtained via grid search from [67]. Specifically these are: 30,000 steps of which 1000 were warmup and linear learning rate decay; batch size 256; one encoder layer and three decoder layer each with a hidden dimension of 1024 and two attention heads; the optimizer was Adam.

**RU-Transformer:** We followed the hyperparameters reported by [12]. These are: 128 dimension hidden size with 256 feedforward; 8 attention heads; 3 layers; batch size 256; trained using Adam with learning rate $10^{-3}$.

## A.9 Compute resources

All reported sDTM runs could be processed on NVIDIA 16gb V100 GPUs. Depending on availability, we ran some seeds on 80gb H100 GPUs, but this is not necessary. The Transformer baselines were also run on NVIDIA 16gb V100 GPUs. NQG used NVIDIA 40gb A100 GPUs. The GPUs we used were hosted on an internal cluster.

Designing our architecture involved many preliminary experiments that are not reported in the paper.

## A.10 Dataset Statistics and Samples

Example input and output pairs are shown for Active↔Logical in Figure 7 FOR2LAM in Figure 8, GeoQuery in Figure 9, and SCAN in Figure 5. The memory usage of DTM grows exponentially with tree depth, so we present the max depth of the datasets here:

- Active↔Logical: max tree depth 10
- FOR2LAM: max tree depth 14
- GeoQuery: max tree depth 16
- SCAN: max tree depth 8

## A.11 Licenses

**Baselines:**

- DTM: Permissive 2.0
- Transformer: BSD-3 (Pytorch implementation)
- RU-Transformer: MIT Licence
- NQG: Apache 2.0

**Datasets:**

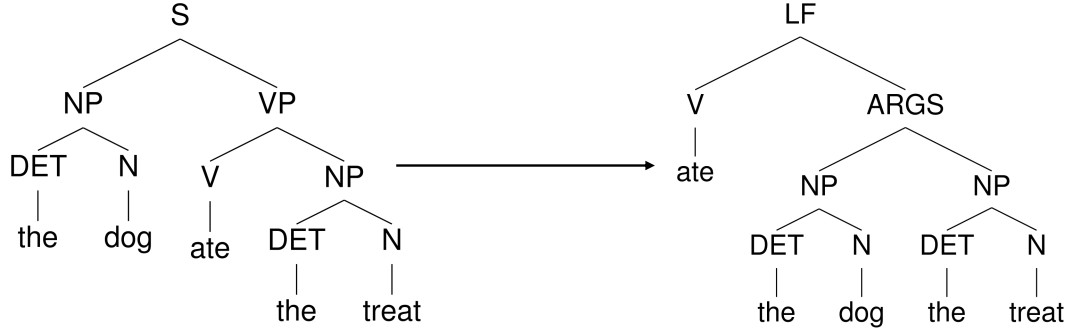

Figure 7: An input and output pair from Active↔Logical.

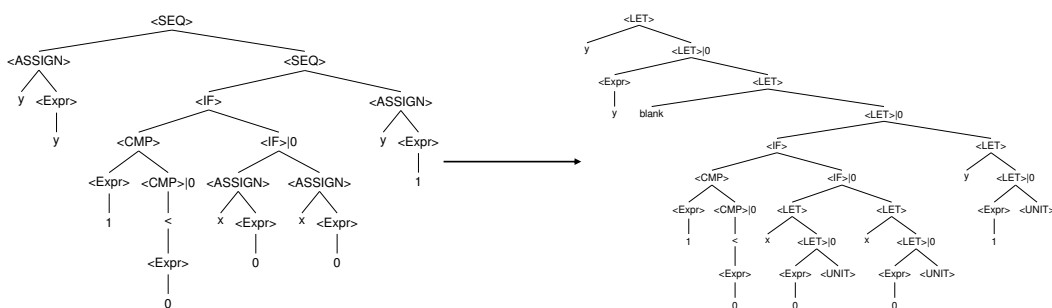

Figure 8: An input and output pair from FOR2LAM.

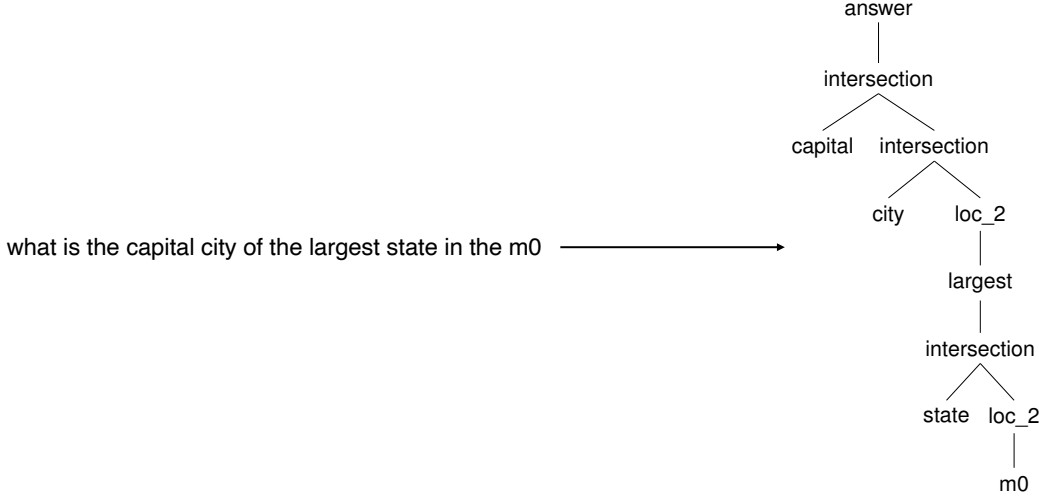

Figure 9: An input and output pair from GeoQuery.

- GeoQuery: GLP 2.0
- SCAN: BSD
- Active↔Logical: Permissive 2.0
- FOR2LAM: Not public (no licence obtained through email request to original authors)

