# OpenReview forum: "Compositional Generalization Across Distributional Shifts with Sparse Tree Operations"
_NeurIPS.cc/2024/Conference — NeurIPS 2024 spotlight_

### Official Review · Reviewer_mbkf · 2024-07-01

**Soundness:** 3
**Presentation:** 4
**Contribution:** 3
**Rating:** 7
**Confidence:** 4

**Summary:**

Authors propose a new representation that they call Sparse Coordinate Trees. When applied to Differentiable Tree Machines, they make computation much more parameter and memory efficient. Due to clever design, the SCTs allow for much more efficient tree operations by bit-shifting, indexing, and addition. Because the tree will naturally become very dense, they apply pruning to make it more sparse. They also propose how to adapt it for sequential inputs and outputs, rather than in tree-form.

In the experimentation section, they provide results on IID, zero-shot, one-shot, structural / length, and template tasks to test generalization, showing that in some ways these methods outperform previous work.

**Strengths:**

A) Professional and clear writing

B) The number of parameters is clearly reduced from the original DTM

C) The memory usage is reasonably reduced, and very reduced for the pruned version

D) Operations are quite a bit more efficient

**Weaknesses:**

A) It would be clearer if there was a better description of what the left, right, and cons functions are intended to accomplish, as this is quite central to the methods

B) There is a lot of extra space in the graphs and 5 runs--adding standard deviation would be nice

C) In Table 1, I am not fully convinced this dataset presents a fitting challenge. The IID are already at 1.0 for almost every method, providing no meaningful distinction between them (although on its own this is maybe fine, as the OOD tasks are the focus). However, both OOD sets go from primarily 0% in the previous works to 100% in this work--not only does it make it seem like the task is very easy once attempted, but it also makes it very difficult once again to differentiate between methods.

D) The only comparison to DTM is in Table 1--aside from lowering resource consumption, it is not clear if the methods have any performance difference (and as pointed out in C, it is not clear if they are truly equivalent or the dataset is just too easy). DTM should really be included in the other experiments. That way, it is clear if sDTM has added performance or just implementation efficiency. If it is just efficiency, then more experiments showing time, memory, parameters, etc would be more fitting than many separate results.

E) Overall, the space is not well used (lots of white space, especially in experiments). It would be better if this were used to showcase more in the paper.

F) Method consistently shows bad performance on MCD (worst of all 4 methods in Table 3, very bad in Table 4)

G) Same as F, but for length experiment

H) It is inconsistent where / when the different tasks are presented. e.g. sDTM seems to be good at 0-shot lexical and structural, but structural is only shown in Table 1. If this is where it is good, it would be much more interesting to see more of that task, than to see MCD in two places (Table 3 + 4), even though sDTM is consistently bad at these tasks.

My primary concerns are
1) I'm not fully convinced on the novelty, because how I understand, it is mostly a more efficient version of DTM, but the experiments focusing on this are very limited and it is also not compared much to DTM in terms of performance (see C, D, E)

2) the results are not great (see C, F, G). In some tasks the method underperforms across the board (length, MCD), and in others where it is good, there is a shortage of results (see H)

**Questions:**

A) in line 306, you say the variants score 0.03%, however in Table 2, they have .03 -- do you mean 3% or .0003? Or are the values in the table truly in percent, and the method is just getting 1%?

B) Do I understand correctly that in Table 3, sDTM performs the worst of all metrics on all 4 tasks?

**Limitations:**

Limitations are adequately addressed.

---

> ### Author Rebuttal · Authors · 2024-08-06
>
> # Reviewer mbkf
>
> Thank you for the time you spent reviewing our paper. Addressing your feedback will make our submission much stronger.
>
> ## Weaknesses
> A. We will make sure to improve the motivation for using left, right, and cons in the camera-ready version. The design of DTM was motivated by the fact that the vast majority of classic symbolic AI was carried out in Lisp, using binary trees as the universal data structure, and using left, right, cons as the complete set of universal structural operators (literally Turing-universal when deployed with the control operation of conditional branching on testing for symbolic equality, as noted in the paper on lines 141-144). This generality means DTM and sDTM are potentially applicable to the huge range of AI tasks previously solved with Lisp.
>
> B. We acknowledge that Table 1 and Table 3 both have whitespace. We will use the whitespace in the tables to include summary statistics about the five runs in the camera-ready version. Can you please let us know where you see excessive whitespace in other figures and tables? Additionally, we experimented with a vertical version of Figure 2 to try and reduce whitespace. The vertical version is available in the PDF attached to our global rebuttal. We would be grateful to know whether you think that this design is a better use of space.
>
> C. We are a bit confused by this comment. Highlighting that some models go from 100\% IID performance to 0\% OOD performance is a signature metric in compositional generalization. In addition to Table 1, we find a similar pattern on FOR2LAM (Table 2), SCAN (Table 4). This pattern is also present in other compositional generalization tasks such as COGS, PCFG, and CFQ. If this observation does not address the issue, we would appreciate a clarification of the problem we should address.
>
> D. This point was also mentioned by other reviewers, and it is clear to us that we did not provide adequate explanation for why we did not test DTM on FOR2LAM, GeoQuery, and SCAN. Please see our response to this in the **sDTM vs DTM** section of the Global Rebuttal.
>
> We appreciate your point that we should emphasise the efficiency gains of sDTM. In addition to parameter and memory efficiency, sDTM is also almost twice as fast as DTM, a point that we did not highlight in the submission. We will correct this in the camera-ready version so that the efficiency improvements of sDTM are better explained.
>
> E. If you care to let us know where you find there is excessive whitespace, we would appreciate it, and would take steps to eliminate it. In order to fit within the NeurIPS style guidelines, there is only so much we can do about whitespace around Tables. We have much more control about whitespace in Figures and are happy to update any figure that you mention as poorly using space.
>
> F. MCD distribution shift is one where we did not find benefits over Transformer and RU-Transformer. We address these limitations in the Conclusion on lines 344-347, where we also highlight that despite the poor performance on MCD shifts, our model performs well in comparison to baselines across the widest variety of distributional shifts.
>
> G. We grouped structural and length generalization together to highlight that generalizing to novel linear positions can also be viewed as generalizing to novel structural positions in a latent parse tree. As shown in Figure 1, our model has the best overall performance on length/structural distributional shifts. One potential area for this confusion is the performance of sDTM on SCAN where the model performs significantly better when the output strings are parsed in a meaningful way.
>
> H. Please see our response to this in the **Layout of the experimental results** section of the Global Rebuttal.
>
> One of the goals of our submission is to highlight distributional shifts and how different models perform differently across them. With this goal in mind, we thought that it was important to include results for sDTM on MCD shifts even though sDTM's performance is disappointing.
>
> ### Primary Concerns
> 1. We failed to explain in our submission that DTM is unable to train on FOR2LAM and GeoQuery. We hope that our explanation in the **sDTM vs DTM** section of the Global Rebuttal adequately highlights the extent to which sDTM provides drastic efficiency improvements. Our response to Weakness H also explains how each experiment is associated with a novel contribution from our work.
> 2. We would like to reiterate that our results illuminate how different models, even ones designed for compositional generalization, perform differently across different datasets and distributional shifts. While NQG has good 1-shot lexical generalization accuracy, we found that it has no 0-shot generalization accuracy. Additionally, while NQG has perfect length generalization on SCAN, it does not outperform a Transformer on the GeoQuery length split and has 0\% accuracy on the Active$\leftrightarrow$Logical structural split. We hope that sDTM’s performance when considered across all of the datasets and splits in our submission highlights our contribution.
>
> ## Questions
> A. Thank you for catching this typo, line 306 should read 3\% and not .03\%. Our model scores 100\% IID and 61\% on 0-shot lexical generalization.
>
> B. Yes, we included results of GeoQuery to show how sDTM and the baselines all struggle on this task. Please see our response to this in the **Performance concerns** section of the Global Rebuttal.
>
> ## General
>
> Please let us know if we did not adequately address any of your questions or concerns. We are committed to engaging with you during the discussion period to continue improving our paper. If you find that our changes in response to your feedback improved our submission, please consider increasing your rating for our paper.$\newline\newline$
> Thank you again for your time and consideration. We look forward to hearing back from you.

---

> > ### Comment · Reviewer_mbkf · 2024-08-08
> > **Response to Rebuttal**
> >
> > Thank you very much for your detailed and thoughtful response, you have addressed many of my concerns. There are just a few points I would like to continue discussing (the points omitted here I am happy with, and would like to thank you for addressing accordingly).
> >
> > WB/E) The updated figures in the vertical format look nice, and look like they should already reduce a bit of space. In terms of additional space-saving suggestions, because it requires some playing to see the actual impacts, I can't be 100% certain if these would work, but some things I could think of to try:
> > - for the updated version of Figure 2, you could also abbreviate Tree positional index (e.g. Tree pos. ind., or TPI with an explanation the caption, or something else fitting), which would allow you to remove quite a bit of white-space on either side of the indices. Then you could either reduce the font size in the top part of the figure or move some of it closer together (maybe The.101 and NP.101, as fun.101 and person.111 are already close)
> > - for Figure 4, the boxes around Agent/Interpreter/Memory are quite high--a bit of space could be saved vertically by shrinking them a bit. This might also look a bit cleaner
> > - All of the tables have a lot of white space between the columns, especially where the titles are long (e.g. 1-shot lexical). If you can shorten/abbreviate them fittingly (an easy example being Length as Len., or with explanations in the captions), you can make them shorter across and either put two on one line or make them in-line with the text, like you did for Table 2. Alternatively, you can use the space to report other information (e.g. higher-level stats, as discussed in WB). Also with Table 3, because it does not take up the full width, a lot of space is wasted to the right and left of the table--you could find another table to put side-by side (e.g. table 2, with a bit of shrinking described above) or also put it in-line with the text.
> >
> > Of course, this space use does not affect the rating directly, but there is some opportunity cost for the information that could have been otherwise presented with this space.
> >
> > WC) You are correct, that the 1.0 IID to 0.0 0-shot lexical is typical / understandable. I'm a bit more concerned with how 0-shot lexical progresses across models--for Transformer, RU-Transformer, and NQG it is 0.0 (again, probably understandable), but for DTM and sDTM it is 1.0. If the first work that addresses the 0-shot lexical case immediately goes from 0% to 100%, is the dataset not perhaps too easy? And it makes it impossible to compare between the different methods, as they all score 100%. In the FOR2LAM dataset, for the 0-shot lexical split this accuracy goes from 0.03 in previous work to 0.61--not only is the dataset sufficiently hard to show that the problem is still unsolved, but if there were other competitors here (e.g. DTM), it is more likely there would be something more meaningful to compare between methods. As a whole, I'm concerned the bar is set too low here in the 0-shot lexical setting for in-depth method comparison.
> >
> > WF) This is fair enough--unfortunate, but you're right that it's better to report the method fairly.
> >
> > Primary concern 1: this makes sense, then I agree that simply being able to apply a DTM-like algorithm in these new scenarios is its own novelty
> >
> > Primary concern 2: with your response in mind, I'll concede that the results are fair. My primary take-aways: exp. 1 shows improved results in 0-shot lexical and structural (as discussed in C, I'm not 100% convinced on the dataset, but either way it is clearly an improvement over previous work); FOR2LAM results are clearly improved; SCAN shows that the 0-shot lexical and template improves over NQG, although the other splits are worse, which is just a trade-off between methods; and GeoQuery results continue to be disappointing showing that the dataset is difficult (but also that sDTM is particularly affected by the difficulties). At the very least, there is a strong case for using sDTM in the 0-shot lexical case. TLDR: I agree there are some clearly compelling use-cases here, even if it is not better across the board.
> >
> > Q2) How you addressed this in the general rebuttal is very nice--if you can turn this section from implying that sDTM doesn't work well (now it is now) to an analysis of how sDTM works / what information it leverages / in what scenarios it is most effective (similar to what you wrote in the general rebuttal), it could turn the section from a weakness into a strength. Saving some white-space in the figures as discussed in WB/E might give you some extra space to go more in-depth into this.

---

> > > ### Author Response · Authors · 2024-08-10
> > >
> > > B/E. Thank you for clarifying your concerns with the whitespace and proposing solutions. We now understand what you are asking for in the other figures and tables and will implement your suggestions. The updated version will reflect this, and we agree that the best use of the extra space this affords us is to provide "an analysis of how sDTM works / what information it leverages / in what scenarios it is most effective".
> > >
> > > C. We understand your concern about Section 5.2 and believe that this may be addressed by further clarifying the purpose of this experiment. While we think your concern about the simplicity of the Active$\leftrightarrow$Logical is well founded, it is one of the main datasets used in the paper introducing the original DTM. As a result we felt it was important to include here in order to show that sDTM with 70x fewer parameters still retains the performance of the original DTM. To make the purpose of the experiment in Section 5.3 clearer, we proposed changing the section header to **Performance Regression (Active$\leftrightarrow$Logical)** in the general rebuttal - and will add a couple of sentences at the start of the section to this effect. As for raising lexical OOD from 0 to 1, this is one of the goals behind making the operations in the programs learned by sDTM blind to lexical identity: the tree operations exploit the the factorization of structure and content to purely manipulate structure, carrying along whatever content (symbols, familiar or novel) may be contained in the structural positions.
> > >
> > > However, your question about the capability differences between DTM and sDTM still stands. Since both DTM and sDTM achieve ceiling performance on Active$\leftrightarrow$Logical, it is impossible to tell what the capability differences are from this experiment. We  think adding DTM results to SCAN will help illuminate the differences between DTM and sDTM, especially since sDTM does not have ceiling performance across all of the splits.
> > >
> > > WF/Primary Concern 1/Primary Concern 2: We are glad that our responses to Weakness F, Primary Concern 1, and Primary Concern 2 were helpful. With regard to the first experiment results discussed in Primary Concern 2, we hope that our discussion on Weakness C above further alleviates your concerns. Please let us know if there is anything that you wish to continuing discussing on those areas.
> > >
> > > Q2: Thank you for your kind words, your feedback was essential in encouraging us to put down our reasoning into words. We will certainly update our discussion of GeoQuery results to include the content in our general rebuttal.

---

> > > > ### Comment · Reviewer_mbkf · 2024-08-12
> > > > **Change of Rating**
> > > >
> > > > Thank you very much for all your thoughtful clarifications. I believe that the changes we've discussed make the paper much stronger, and the discussion in general has made the paper much clearer to me.
> > > >
> > > > I am raising my score to Accept, for the following reasons:
> > > > 1) I see that I was overly harsh on very specific aspects of the experiments in my first rating. Overall, there are compelling use-cases here.
> > > > 2) The storyline through the experimental section has greatly improved, making the benefits of the model much more clear.
> > > > 3) The novelty is much clearer now (especially with respect to DTM).
> > > >
> > > > I have a hard time raising the score higher, because to me the results are not improved enough across the board to justify a higher rating. However, I fully believe the paper should be accepted.

---

> > > > > ### Author Response · Authors · 2024-08-13
> > > > >
> > > > > Thank you for the fruitful back-and-forth during the discussion period! The changes we arrived at will make the paper much clearer to readers, especially our novel contributions and storyline through the experimental section. We understand your current rating and hope that future work will allow us to further increase performance across more distributional shifts and larger datasets.

---

### Official Review · Reviewer_Hmrt · 2024-07-11

**Soundness:** 3
**Presentation:** 4
**Contribution:** 2
**Rating:** 7
**Confidence:** 4

**Summary:**

The paper proposes a novel way of representing sparse trees where nodes have vector attributes in a denser, tensorised format which they call Sparse Coordinate Trees (SCT). Essentially, the crucial component for SCTs is to represent the indices of the nodes according to their topological ordering, allowing for all nodes to be represented by a vector of indices and a tensor of attributes. Additionally, the authors also show how some traditional operations on trees, such as taking left/right subtrees or constructing a new tree from left/right branches, can be efficiently implemented with simple indexing or bitshifts when working on the binary representation of the node indices. Moreover, these operations can be parametrised in a differentiable way using modern machine learning models, such as transformers, opening the door to learning the structure of a SCT

The next contribution is then to use SCT to extend an existing neurosymbolic [1, 2] framework called Differentiable Tree machines (DTM) to be able to work with sequence data (seq2tree and seq2seq tasks) instead of just tree data (tree2tree tasks). While maintaining the semantics of DTM, the use of SCT as an inference engine is also shown to be more memory and parameter efficient. Finally, the theoretical claims of the paper are supported by a strong suite of four benchmarks.

[1] Garcez, A. D. A., & Lamb, L. C. (2023). Neurosymbolic AI: The 3 rd wave. Artificial Intelligence Review, 56(11), 12387-12406.

[2] Marra, G., Dumančić, S., Manhaeve, R., & De Raedt, L. (2024). From statistical relational to neurosymbolic artificial intelligence: A survey. Artificial Intelligence, 104062.

**Strengths:**

1. The paper is very well written and easy to follow. The motivation for the ideas in the paper and their explanations are clear.

2. The suite of experiments is quite extensive, covering the three kinds of tasks discussed in the paper (tree2tree, seq2tree, seq2seq) on recognised datasets. Apart from the number of experiments, the advantages of sDTM compared to the chosen baselines are also clear in most cases.

3. The idea of the paper is simple, but elegant and it is easy to see why it can give substantial improvements in terms of efficiency. It also nicely allows for the incorporation of modern machine learning models like transformers. While I am unsure about the overall impact of the work as it seems there are many questions left to answer, the questions and answers about generalising beyond mere i.i.d. training and test cases are crucial and tie in with the current rise of neurosymbolic AI.

**Weaknesses:**

While I overall enjoyed reading this interesting paper and appreciate the provided insights, I do have some comments and questions:

1. A lot of related work is properly and extensively discussed in Section 2, yet I do believe a series of references might be missing. There are many more general neurosymbolic frameworks that use neural nets to parametrise symbolic components. Some are based on fuzzy logic [1], while others use probabilistic logic [2, 3], in contrast to being based on tree structures. For the special case of sequences, a system based on stochastic grammars also exists [4]. Even more, some of these systems go further in allowing neural embeddings to be present within the logical system [5, 6], similarly to how nodes in the tree are composed of their vectors of attributes. It is true that many of these systems have not focused on structure learning and do assume some prior knowledge, which is not a prerequisite for the proposed method. Although this area of "structure learning" (or learning a *program* as line 157 puts it) is an active area of research [7, 8]

2. In section 3.2 it is shown that the operations *left*, *right* and *cons* can be implemented very efficiently as tensor operations. However, not much is said about the additional operations of *conditional branching* and *equality-testing*. It is mentioned that the five operations together are Turing complete, but only the first three seem to be used and nothing is said about an implementation of the last two for SCT.

3. sDTM extends DTM to tasks different from just tree2tree tasks, but it is not completely clear how much of this is due to the use of SCT. For example, to allow even sDTM to deal with seq2seq tasks, the authors do need a hardcoded translation from output trees to sequences. I wonder if a similar hardcoding could not have been used for input sequences to trees, allowing vanilla DTM to deal with the same coverage of tasks, albeit with a less flexible input encoding.

4. Most of the experimental results are promising, but some results did raise some questions (see below in the questions section). It would also be nice to see some examples of some of the datasets, even if only in the appendix, to make it more tangible what the input and output is of the experimental tasks.

5. As briefly mentioned previously, the true impact of this work remains hard to guess. There is surely a lot of promise in neurosymbolic methods in general and the proposed SCTs and sDTM do show improved generalisation performance by learning both neural and symbolic components *from scratch* and *from data*. However, the use of *only* tree structures could prove limiting for applications with more intricate dependencies.

Smaller concerns:

1. Section 4.3 talks about how pruning can be used by keeping the top-$k$ nodes. However, it is unclear whether this can be done during training, since the top-$k$ operation is not differentiable.

2. On line 197 it is stated that attention is permutation invariant, yet this is not completely correct. Attention is invariant to permutations of the keys and values and only *equivariant* to permutations of the queries. It would be good to make the distinction clear to avoid confusion.

[1] Badreddine, S., Garcez, A. D. A., Serafini, L., & Spranger, M. (2022). Logic tensor networks. Artificial Intelligence, 303, 103649.

[2] Yang, Z., Ishay, A., & Lee, J. (2020, July). NeurASP: Embracing Neural Networks into Answer Set Programming. In 29th International Joint Conference on Artificial Intelligence (IJCAI 2020).

[3] De Smet, L., Dos Martires, P. Z., Manhaeve, R., Marra, G., Kimmig, A., & De Readt, L. (2023, July). Neural probabilistic logic programming in discrete-continuous domains. In Uncertainty in Artificial Intelligence (pp. 529-538). PMLR.

[4] Winters, T., Marra, G., Manhaeve, R., & De Raedt, L. (2022, June). Deepstochlog: Neural stochastic logic programming. In Proceedings of the AAAI Conference on Artificial Intelligence (Vol. 36, No. 9, pp. 10090-10100).

[5] Rocktäschel, T., & Riedel, S. (2016, June). Learning knowledge base inference with neural theorem provers. In Proceedings of the 5th workshop on automated knowledge base construction (pp. 45-50).

[6] Maene, J., & De Raedt, L. (2024). Soft-unification in deep probabilistic logic. Advances in Neural Information Processing Systems, 36.

[7] Shindo, H., Nishino, M., & Yamamoto, A. (2021, May). Differentiable inductive logic programming for structured examples. In Proceedings of the AAAI Conference on Artificial Intelligence (Vol. 35, No. 6, pp. 5034-5041).

[8] Muggleton, S. (1991). Inductive logic programming. New generation computing, 8, 295-318.

**Questions:**

Apart from the concerns raised in the previous section, here are a couple more specific questions:

1. What is the intuition behind only using a single learnable parameter for the query vector? (lines 205-206)

2. While it is nice to see that sDTM generally does perform better than transformers in terms of OOD generalisation, I am left wondering if it also can not be prone to the same pitfalls as transformers as SCT and hence sDTM do utilise transformers internally to construct trees. Could you elaborate on this as it could be an important limitation of this paper? As experimental support for this limitation, the lacking performance in the experiment of Section 5.4 could be evidence. Additionally, the imperfect score of 0.61 in Table 2 can also be seen as evidence for this, given the rather simple nature of "replacing the name of a variable".

3. Experimental questions:
     + Why is the original DTM only present in Table 1 and not Table 2? Both relate to tree2tree tasks where DTM should also be applicable if I understand correctly.
     + Lines 283-285: How much of the 20% memory reduction is due to the use of pooling by attention and how much is due to using SCT?
     + Table 2: sDTM gets a score of 0.61 on the 0-shot test set where one variable name is consistently changed to another. Do all test set occurences contain the variable x that is changed to z?
     + In general, the evaluation metrics should be explained a bit more in detail. For example, what does it mean for a FOR2Lam translation to be correct? The translated AST is exactly the same as the target, or equivalent in some way?
     + Lines 321-322: the small dataset is used as argument for the lacking performance of sDTM compared to other methods, but do those other methods not also suffer from the small dataset? Transformers are known to be rather data-hungry, so I would still have expected sDTM to outperform them at least in this task. Do you have some deeper intuitions as to why this is not the case?
     + In general, why the choice of the best performance out of 5 runs? While means and standard deviations are certainly not always ideal, aggregate and variability metrics are still more insightful to gauge the consistency of the tested methods. If one does not want to use means and standard deviations/errors because of their distributional assumptions, medians and quantiles are a good solution.
     + It seems like there are more neurosymbolic methods that could be applied to the discussed tasks, such as those mentioned in the related work in lines 97-98. Why the choice for only NQG?

4. I am curious about the overall training times for all methods, to see if sDTM requires substantially more time to train or not. Can you comment on this please?

In general, I do give a more positive rating to this paper as its presentation is excellent, its contribution is interesting and its experimental evidence is quite convincing. I will gladly further increase my score if the authors can answer my concerns.

**Limitations:**

I believe limitations are sufficiently addressed, as the conclusion specifically mentions that sDTM still struggles with some OOD generalisation tasks. However, some potentially limiting factors, such as training times, are not immediately clear.

---

> ### Author Rebuttal · Authors · 2024-08-06
>
> # Reviewer Hmrt
> Thank you for the extensive feedback you provided on our submission! Addressing your questions and the weaknesses you identified will significantly strengthen our work. We are pleased that you found our work “very well written”, “the suite of experiments is quite extensive”, and “the idea of the paper is simple, but elegant”. Below we will address the weaknesses and questions that you identified.
>
> ## Weaknesses
> 1. We are grateful for the additional references that you provided. We will amend the Related Work section in the camera-ready version to include a subsection on Neurosymbolic Computation with the references you provided, as well as additional citations.
> 2. Thank you for catching this oversight. Conditional branching and equality-testing are the control flow mechanisms for how to sequence the structural operations as well as argument selection. In this regard, conditional branching and equality-testing are implicitly parameterized by the Agent using a Transformer. In numerous approaches to neural program synthesis, neural networks are used to parameterize program control flow (e.g. Nye et al (2020), where networks generate symbolic rule sequences). Unlike most of that work, however, in our approach the synthesized programs are themselves differentiable. We will update the camera-ready version to make this contribution of our work clear.
>
> 3. Please see our response to this in the **sDTM vs DTM** section of the Global Rebuttal.
>
> 4. Thank you for this suggestion. We included an example sample from SCAN in Figure 5 and will add addition samples across the datasets to the Appendix in the camera-ready version.
>
> 5. Extending data structures in superposition to graphs is an active area of research. We focused on trees given their importance in language processing, but we would like to generalize the techniques in our paper to graphs as well. Many data structures such as lists, stacks, and queues can be implemented using left, right, and cons, so it is possible for sDTM to succeed on tasks that would be better modeled by these data structures. This is an interesting point that we intend to investigate in follow-up work.
>
> Smaller Concern 1: top-k selection is also done during training. Top-k selection can be viewed similarly to deterministic channel-wise dropout. As such, when a node is dropped, the node does not contribute to the output, and thus it will have a gradient of 0. We will include additional text in the camera-ready version to make sure that it is clear how top-k selection works with regards to training.
>
> Smaller concern 2: this is a good point, thank you for your attention to detail! We will update the text to make it clear that attention is permutation equivariant.
>
> ## Questions
> 1. The sentence that you pointed out is written incorrectly. We have n_head query vectors each of dimension d_key. We will correct this in the camera-ready version. Thank you for spotting this typo.
> 2. Good point, and this ties in to point 2 you raised in the Weaknesses section. As long as the Transformer can make the correct control flow predictions, what operations to perform on what trees in memory, the transformation will have good generalization. However, sDTM can still fail to generalize correctly when the underlying Transformer that powers the agent does not make correct control flow predictions. We address some concerns about model performance in the **Performance concerns** section of the Global Rebuttal.
>
> 3. Experimental questions
>     1. Please see our response to this in the **sDTM vs DTM** section of the Global Rebuttal.
>     2. This is a good question, and we don't have a definitive answer just yet. SCT only provides memory savings when nodes are empty. DTM with more layers will eventually lead to fully dense trees, in which case all of the memory savings will be due to pooling by attention. However, in this specific example, the early sparse trees will account for some of the memory savings.
>     3. Yes, every test set sample contains a variable x that is changed to z in the 0-shot test set.
>     4. The evaluation metric that we use is Exact Match Accuracy. We will update the paper to clarify this.
>     5. Please see our response to this in the **Performance concerns** section of the Global Rebuttal
>     6. We find that sDTM is prone to getting stuck in local optima and followed previous papers in reporting the best run, as described on lines 982-986 in the Appendix. We acknowledge that summary statistics are still helpful. We want to continue reporting the best performance since this highlights the theoretical capabilities of our model, but we will also include summary statistics in the updated version.
>     7. We picked NQG as our signature hybrid neurosymbolic model since it was applied to different distributional shifts and datasets. Most other hybrid neurosymbolic models are heavily tied to a specific dataset or distribution shift, and we did not have the resources to adapt models for each task and distribution shift.
> 4. Thank you for pointing out this oversight, we should have included information about the training time for each architecture and will rectify this. On Active$\leftrightarrow$Logical, 20k steps took DTM 11 hours, sDTM 6 hours, and our Transformer 2 hours. While the forward and backward speed of Transformer is still much faster than sDTM, it uses highly optimized CUDA kernels, whereas sDTM has a lot of room for improvement.
>
> ## General
>
> Please let us know if you have any additional questions or concerns. If we addressed your questions and concerns, please consider increasing your score. We truly appreciate the time that it must have taken to provide such a careful analysis of our submission. Thank you!

---

> > ### Comment · Reviewer_Hmrt · 2024-08-11
> > **Acknowledgement of author rebuttal**
> >
> > Thank you for the extensive answers to my questions and concerns! I sincerely appreciate the honesty in your answers, for example mentioning that sDTM can be prone to getting stuck in local optima. I hope to find this observation together with the other remarks (such as training times and the impracticality of DTM on the FOR2LAM and GeoQuery tasks) in the camera-ready version of the paper. Additionally, I am also looking forward to the additional comparison between DTM and sDTM on the SCAN task.
> >
> > I believe the clarification of using transformers as the overall control flow mechanism, leading to differentiable synthesised programs, will further improve the overall exposition as it shows which parts are symbolic and which parts are neurally parametrised. Especially since readers can then easily identify why sDTM might struggle with OOD generalisation. For example, in cases where the burden of generalisation falls on the neural component.
> >
> > With respect to my first smaller concern, I now see that top-k in the context of the paper indeed can be seen as a differentiable deterministic dropout. It is only when probabilistic semantics are attached to the predictions, e.g. the predicted values are to be interpreted as probabilities, that top-k introduces complications for gradients. Specifically, the same deterministic dropout interpretation would be a biased estimate of the true gradient in case of probabilistic semantics. However, since no probabilistic semantics are being claimed, I agree with the provided answer.
> >
> > I consider my concerns addressed and up my score to a full accept.

---

> > > ### Author Response · Authors · 2024-08-13
> > >
> > > Thank you for your very comprehensive analysis of our submission! Addressing your concerns and questions greatly improved our paper, and the camera-ready version will benefit greatly from these changes.

---

### Official Review · Reviewer_bgti · 2024-07-11

**Soundness:** 3
**Presentation:** 3
**Contribution:** 3
**Rating:** 7
**Confidence:** 3

**Summary:**

This work addresses the problem of compositional generalization in the domain of natural language processing. The authors highlight that incorporating tree structures into a models representation space is important for achieving compositional generalization. To this end, the authors build upon a recent method for incorporating such structure by extending the method such that it is 1. significantly more parameter/memory efficient and 2. able to handle seq2seq task opposed to just tree to tree task. The authors test their method on various natural language task and show superior compositional generalization performance across several metrics relative to baseline methods as well as improved efficiency relative to the method they build upon.

**Strengths:**

* This paper addresses an important problem; namely, closing the gap between human's and machine's ability to generalize compositionally in natural language task.

&NewLine;

* The paper is very well written, well structured, and easy to understand.

&NewLine;

* Section 2 provides a solid review of prior works and does a good job contextualizing the authors contribution relative to prior works.

&NewLine;

* The authors method yields promising empirical results both in terms of memory efficieny and performance relative to existing baselines e.g. Transformers.

&NewLine;

* The authors are upfront and transparent when their method underperforms in Section 5.4 and aim to provide potential explanations for why this may be occurring.

**Weaknesses:**

__1.__ I found the experiments section to be a bit unclear in its focus. As I understand, one of the core points of sDTM and thus the paper, is that sDTM is significantly more memory/compute efficiency than DTM. While the authors compare these two methods in terms of efficiency and performance in 5.2, such an experiment does not exist for the tree2tree task in 5.3. Consequently, I am a bit confused on the purpose of the experiments in 5.3 given the main message of this work.

&NewLine;

Specifically, the current point seems to be to show that sDTM outperforms baseline methods on FOR2LAM. Given that sDTM is an extension of DTM, the scores for sDTM in isolation do not seem particularly meaningful without a relative comparison to DTM as was done in 5.2. Please let me know, however, if there is something I am missing here.

&NewLine;

For the same reasons, I think the experiments in 5.4 and 5.5 would also benefit from reporting scores for DTM alongside sDTM. I suppose the original DTM method cannot be directly applied since these inputs involve sequences, however, if I understand correctly, it seems the same technique to deal with sequence task in sDTM can be applied to DTM.

&NewLine;

$\newline$

&NewLine;

__2.__ It would be important to understand the compositional generalization benefits obtained by sDTM over e.g. Transformers on more large scale models/datasets, however, I also recognize that such a study could be out of the scope of this work.

&NewLine;

$\newline$

&NewLine;

__3.__ There are some cases in which sDTM does not offer benefits over existing baselines as reported in Section 5.4, however, the authors are upfront about this in the paper.

**Questions:**

* What is the primary purpose of the experiments in Sections 5.3-5.5?

&NewLine;

* Do the authors have intuition for how well sDTM could scale to more complex models/datasets? In particular, given its compute efficiency over DTM?

&NewLine;

* Do the authors envision that sDTM could be applied in task outside of natural language, e.g. visual reasoning or planning task?

**Limitations:**

This paper does not contain an explicit limitations section, however, the authors provide a transparent discussion about some limitations of sDTM in Section 5 and in the Conclusion section.

---

> ### Author Rebuttal · Authors · 2024-08-06
>
> # Reviewer bgti
> Thank you for the time you spent to understand our submission and provide valuable feedback! Addressing your concerns will make our paper substantially stronger. We are pleased that you found that our paper “addresses an important problem”, is “very well written”, contextualizes our contribution in relation to prior work, and “yields promising empirical results”. Below we will address the weaknesses and questions that you highlighted.
>
> ## Weakness
> 1. Our original submission did not adequately address the significance of each experiment, and we failed to explain when it is possible to compare DTM and sDTM. As you and other reviewers point out, we only directly compare sDTM and DTM in Section 5.2. The reason that we did not include DTM in Section 5.3, and this is a point we will be sure to highlight in the camera-ready version, is that DTM cannot fit a batch size of 1 into memory on FOR2LAM. Theoretically, DTM should be able to solve FOR2LAM, but the inefficiencies with regard to exponential growth due to tree depth (lines 111-115) make it impossible in practice. The purpose of Section 5.3 is to show that sDTM, which has the same theoretical guarantees as DTM but is practically more efficient, can scale up and solve FOR2LAM.
> \
> \
> Your understanding of the separation between our contribution of seq2tree and seq2seq and architectural improvements in sDTM is correct. The same technique to process sequences that we introduce alongside sDTM can also be applied to DTM. GeoQuery has deeper trees than FOR2LAM, which means that it is also not feasible to test DTM on GeoQuery. However, it is practical to train both sDTM and DTM on SCAN. We will run the experiment to test DTM on SCAN and will update the camera-ready version with these results.
> \
> \
> In order to make the purpose of each experiment clear, we propose to change the subheaders in the Results section of the camera-ready version to better reflect the significance of each experiment, with the associated dataset in parentheses. **Section 5.2: Performance Regression (Active$\leftrightarrow$Logical)** confirms that sDTM does not worsen DTM's original performance. **Section 5.3: Scalability (FOR2LAM)** investigates a tree2tree transformation task that DTM cannot handle as explained in the previous section of this rebuttal. **Section 5.3: Seq2Tree (GeoQuery)** introduces the change of processing a sequence as input instead of a tree, and **Section 5.4: Seq2Seq (SCAN)** adds an additional modification of sequence outputs.
>
> 2. We hope that our more efficient implementation of DTM will allow us in future work to compare the compositional generalization benefits of sDTM and Transformers on larger datasets and models. In this work, we focused on making it practical to test DTM on a wider variety of tasks by making it much more efficient and capable of processing sequence inputs and outputs. As mentioned in the previous section, the original DTM could not be tested on FOR2LAM and GeoQuery because it was too inefficient to fit a batch size of 1 into the available memory. sDTM provides a base for us to approach larger datasets in future work.
>
> 3.  While we would be excited if sDTM was the best model across every split and every dataset, Figure 1 shows that sDTM's improvement over baselines comes when results are considered as a whole. As we note in our general response to reviewers, we aim to present results across a breadth of distributional shifts to help shed light on what kinds of generalization different architectures excel at.
>
> ## Questions
> 1. Please see our response to Weakness #1 above.
> 2. Please see our response to Weakness #2 above.
> 3. With regard to other modalities, we hope that sDTM can be applied to any task that can benefit from hierarchical representation and processing. As you mention, many types visual reasoning and task planning, such as scene understanding and goal decomposition into subgoals, can be formulated as hierarchical problems. The challenge here will be going from raw input such as an image into a tree-structured representation, as well as managing this encoding in a way that generalizes.
>
> ## General
> Please let us know if you think that our submission would benefit from an explicit Limitations section.
> \
> \
> We hope that we have addressed your questions and concerns. If you have any additional questions, or we did not address any of your original points, please let us know. We are committed to continuously improving our paper and appreciate your feedback. If you find that our changes in response to your feedback improved our submission, please consider increasing your support for our paper.
> \
> \
> Thank you again for your time and consideration. We look forward to hearing back from you.

---

> > ### Comment · Reviewer_bgti · 2024-08-11
> > **Re: Rebuttal**
> >
> > I thank the authors for addressing my points in detail. I have decided to increase my score to accept.

---

> > > ### Author Response · Authors · 2024-08-12
> > >
> > > Thank you very much for your feedback! We really appreciate the time you spent to help us make our submission stronger.

---

### Author Rebuttal · Authors · 2024-08-06

# Global Rebuttal
We thank all of our reviewers for their careful analysis of our work. In this global response, we highlight points shared by multiple reviewers.

First, we are excited by the kind comments that reviewers provided concerning our paper. All three reviewers found our paper to be clear, easy to understand, and well written. They also highlighted the empirical performance and computing efficiency benefits of our work compared to the original DTM, especially in terms of parameter count and memory usage. We realized that in addition to the benefits that the reviewers highlighted, we failed to describe the speed benefits of sDTM over DTM. Due to the much smaller number of parameters and activation dimensions, sDTM is also much faster than DTM. On Active$\leftrightarrow$Logical, DTM takes 11 hours to to train for 20k steps, whereas sDTM takes 6 hours. We will update the camera-ready version of our paper to include empirical results showcasing the speed improvements of sDTM vs DTM.

The primary feedback across reviewers was the need for greater clarity about the relationship between the original DTM and our proposed sDTM including a more thorough comparison of the architectures. Reviewers also asked for clarification with regard to the importance of each experiment in our Results section. In the sections below, we address these two issues and explain the changes we will make to our work for the camera-ready version informed by your feedback.

## sDTM vs DTM
We agree with reviewers that since sDTM is introduced as a more efficient version of DTM, full results comparing these two techniques are essential. It was an oversight not to include more direct comparisons of sDTM and DTM, and to explain why such a comparison is not possible on certain datasets due to the limitations of the original DTM.

In addition to the representational change (Sparse Coordinate Trees Section 3) and architectural change (pooling by attention Section 4.2) to go from DTM to sDTM, we also contributed an orthogonal technique to process sequence inputs and outputs (Section 4.5). We did not adequately isolate these orthogonal contributions by comparing sDTM and DTM on sequence inputs and outputs. As the reviewers pointed out, only Section 5.2 (Active$\leftrightarrow$Logical) contains a comparison between the original DTM and our proposed sDTM. In the camera-ready version, we will also report results for the original DTM on the seq2seq task SCAN (Section 5.5).

We did not test the original DTM in Section 5.3 (FOR2LAM) and 5.4 (GeoQuery) as the original DTM is so inefficient as to be impossible to run on our available hardware; a batch of only a single sample from these datasets causes an out-of-memory error on a 16gb V100 GPU. To put this in perspective, sDTM is able to process batches of 64 samples on the same GPUs, which exemplifies the dramatic efficiency benefits of sDTM. DTM cannot practically be run on FOR2LAM and GeoQuery because of the max tree depth of samples in these datasets. As explained on lines 111-115, DTM's memory requirement grows exponentially with the max tree depth and quickly runs into performance issues. The maximum tree depth for Active$\leftrightarrow$Logical, SCAN, FOR2LAM and GeoQuery are respectively 10, 8, 14, and 16. We will update the camera-ready version of our paper to explain the absence of DTM results on FOR2LAM and GeoQuery, as well as to include a table in the Appendix with tree depth statistics for all the datasets.

## Layout of the experimental results
Reviewers also sought clarity in how each individual dataset and experiment contributes to our overall contributions. To make the relationship between each experiment and our contributions more clear, we will change the subheaders in the Results section of the camera-ready version to better reflect the significance of each experiment, with the associated dataset in parentheses. **Section 5.2: Performance Regression (Active$\leftrightarrow$Logical)** confirms that sDTM does not perform worse than DTM. **Section 5.3: Scalability (FOR2LAM)** investigates a tree2tree transformation task that DTM cannot handle as explained in the previous section of this rebuttal. **Section 5.3: Seq2Tree (GeoQuery)** introduces the change of processing a sequence as input instead of a tree, and **Section 5.4: Seq2Seq (SCAN)** adds an additional modification of sequence outputs.

## Performance concerns
Multiple reviewers pointed out that sDTM does not achieve state-of-the-art performance across all tasks, with relatively weak performance on GeoQuery (Section 5.4). While we would be excited if sDTM was the best model across every split and every dataset, we want to remind reviewers that the results should be considered as a whole, as exemplified by Figure 1.

It is worth noting that there is substantial room for improvement across every model on GeoQuery. (s)DTM is proposed to complement generic similarity-based generalization (already offered by Transformers) with compositional-structure-based generalization. In tasks lacking sufficient opportunities for compositional generalization, DTM will have limited value to augment generic transformer-style generalization. It appears that GeoQuery is such a task, because the strongly compositional symbolic methods of NQG fail. It is possible that with sufficient data, GeoQuery's latent compositional structure could be identified by NQG and DTM, but the released GeoQuery dataset has only on the order of 500 training examples. Given all methods perform well below ceiling on GeoQuery (including on the IID split), we refrain from drawing substantive conclusions based on minor differences in accuracy on this single task in isolation from the rest of our results. We will update the text to clarify DTM's performance on GeoQuery.

We thank the reviewers again for the time that they dedicated to improving our submission. By responding to their feedback, we feel that our paper is much stronger and understandable.

---

> ### Author Response · Authors · 2024-08-13
>
> We thank all three reviewers for their constructive feedback. The original reviews and discussion period were very helpful in strengthening our submission. We are particularly grateful for the time that the reviewers spent to understand our work; the reviewers' questions and concerns were highly actionable and showed a deep understanding of our work that must have required a thorough analysis.

---

### Decision · Program_Chairs · 2024-09-25

**Decision:**

Accept (spotlight)

**Comment:**

The paper addresses a relevant and significant problem, i.e. how to improve OOD performances in a neural network (in this case in the NLP domain) mainly exploiting the compositional property of symbolic systems. In fact, the combination of neural and symbolic systems (Neurosymbolic Systems) have been studied since the 90', looking for the ``right'' way to combine these two technologies. This paper contributes to the advancement in this field by introducing a new technique for representing trees in vector space called Sparse Coordinate Trees, enabling the definition of structural operations useful for efficiently handling seq2seq task.
Especially after rebuttal, all reviewers agreed that the contribution of the paper is technically solid and significant. The authors are encouraged to embed in the final version of the paper all reviewers recommendations.